:ᐧᐧᐧ: PLOS | ONE

# Barriers for tuberculosis case finding in Southwest Ethiopia: A qualitative study

Berhane Megerssa Ereso[1¤]*, Solomon Abebe Yimer[2,3], Christoph Gradmann[1], Mette Sagbakken[4]

**1** Department of Community Medicine and Global Health, Institute of Health and Society, University of Oslo, Oslo, Norway, **2** Department of Microbiology, Institute of Clinical Medicine, University of Oslo, Oslo, Norway, **3** Coalition for Epidemic Preparedness Innovations (CEPI), Oslo, Norway, **4** Department of Nursing and Health Promotion, Faculty of Health Sciences, Oslo Metropolitan University, Oslo, Norway

¤ Current Address: Department of Health Policy and Management, Faculty of Public Health, Institute of Health, Jimma University, Jimma, Ethiopia
* berhanemegerssa2004@gmail.com

**Data Availability Statement:** All data analyzed during the current study are within the paper and its Supporting Information files.

**Funding:** This study was not funded by a grant. It is a PhD project and was supported by the

## Abstract

### Background

Ethiopia is one of the countries with a high burden of tuberculosis (TB). Jimma Zone has the lowest TB case notification rate compared to the national and World Health Organization's (WHO) targets. The aim of the present study was to identify barriers, and explore the origin of these barriers in relation to TB case finding.

### Methods

A qualitative study was conducted by using different data collection methods and sources. Sixty in-depth interviews with TB treatment providers, program managers and TB patients were included. In addition, 42 governmental health facilities were observed for availability of resources. Data obtained from the in-depth interviews were transcribed, coded, categorized and thematized. Atlas.ti version 7.1 software was used for the data coding and categorizing.

### Results

Inadequate resources for TB case finding, such as a shortage of health-care providers, inadequate basic infrastructure, and inadequate diagnostic equipment and supplies, as well as limited access to TB diagnostic services such as an absence of nearby health facilities providing TB diagnostic services and health system delays in the diagnostic process, were identified as barriers for TB case finding. We identified the absence of trained laboratory professionals in 11, the absence of clean water supply in 13 and the electricity in seven health facilities. Furthermore, we found that difficult topography, the absence of proper roads, an inadequate collaboration with other sectors (such as education), a turnover of laboratory professionals, and a low community mobilization, as the origin of some of these barriers.

Strategic and Collaborative Capacity Development in Ethiopia and Africa (SACCADE) Project, Norwegian Program for Capacity Development in Higher Education and Research for Development (NORHAD), University of Oslo.

**Competing interests:** There are no financial or non-financial competing interests among the authors. The authors have declared that no competing interests exist.

## Conclusion

Inadequate resources for TB case finding, and a limited access to diagnostic services, were major challenges affecting TB case finding. The optimal application of the directly observed treatment short course (Stop TB) strategy is crucial to increase the current low TB case notification rate. Practical strategies need to be designed to attract and retain health professionals in the health system.

## 1. Introduction

Tuberculosis (TB) is a major public health problem in the world [1]. Globally, it is one of the leading causes of death from an infectious disease [2], with an estimated 10.4 million TB cases and 1.6 million deaths from TB in 2016 [1]. Most of the deaths are preventable, as long as people can receive an early diagnosis and proper treatment. Additionally, over 95% of all deaths from TB occur in low- and middle-income countries [3].

Ethiopia is one of the 30 highest TB burden countries (HBCs) in the world. These HBCs account for approximately 87% of the total Global TB burden [2]. Moreover, TB ranks sixth out of the top 10 causes of mortality in Ethiopia [4]. In 2015, the national case notification rate for all forms of TB was 67.3%, with a cure rate of 77.9%. In the Oromia Region in Ethiopia, where this study was conducted, the TB case notification rate was 65.4%. Unfortunately, this achievement is lower than the WHO targets for TB control of a 70% case notification rate and 85% cure rate [4].

The early diagnosis of TB is crucial for controlling TB since this will reduce the length of delay in the diagnosis of TB and prevent transmission. To be able to facilitate the early diagnosis of TB, accessibility to TB diagnostic and treatment facilities is crucial [5,6]. The Stop TB strategy was launched by the WHO and adopted by Ethiopia in 2006 [2,7]. It focuses on pursuing a high quality directly observed treatment short course (DOTS) expansion and enhancement, improving case finding and cures through an effective patient-centered approach aiming at reaching all patients [3]. The continuation of the Stop TB strategy is the End TB Strategy, which covers the period from 2016 to 2035. It has a target of a 90% reduction in TB mortality and an 80% reduction in TB incidence by 2030 from the 2015 baseline [2]. The early diagnosis of TB, and the screening of contacts and high-risk groups in a systematic way, are components of the End TB strategy [1].

The aim of early diagnosis and treatment is to cure more patients, interrupt transmission to healthy individuals, and prevent drug-resistant TB from arising. However, these aims are not yet achieved as expected in many sub-Saharan African countries [2]. Factors that negatively affect early TB diagnosis are limited access to a health facility, and a lack of diagnostic supplies [8–10]. Expenses related to hospitalization, transport, diagnostic investigations and care are other challenges for patients in sub-Saharan Africa [11]. Furthermore, self-treatment, limited community awareness of the risks associated with delayed care seeking and low educational status are all factors significantly associated with delayed TB care seeking. Among those who have not had previous TB treatment, and who have not had a cough for a long duration, there is a delay in TB care seeking [12–16].

According to the Roadmap (strategic document) for TB operational research conducted in Ethiopia, the major problems of the TB control programs in Ethiopia are primarily related to a lack of efficient and effective ways of delivering services and equitably reaching the population at risk [17]. The National TB control program in Ethiopia recently started to scale up

community TB care, using health extension workers to ensure access to DOTS at the kebele level (the lowest administrative unit in the community). In Ethiopia, community-based TB care is provided by health extension workers. These workers provide TB treatment for patients at health posts or at the patient's home, and refer people who screen positive for TB to health facilities [18].

According to a study in Ethiopia, community-based DOTS could enhance existing health services by improving the access to- and success rate of TB programs [19]. Among others, studies have shown that community-based TB care may improve the speed of case findings of smear-positive TB patients. Moreover, it may increase the TB case notification rate (the number of all TB cases notified to the national health authorities per 100,000 population during a specified period of time) and treatment success rate (percentage of all TB cases that successfully completed treatment), while decreasing the loss to follow-up of those on treatment [20,21]. Community-based TB care may also be more acceptable by patients and providers because of its accessibility to a majority of people. It can also make the diagnoses and treatment of TB services more accessible for those who cannot afford costs related to transportation and investigations, and for the elderly [20–22]. Furthermore, it is cost effective due to reduced travel distance, reduced transportation costs and time lost to obtain the services [19].

In Ethiopia, presumptive TB cases are managed using two approaches: (1) For clients with a low risk to drug resistance (DR) TB, HIV-negative and adults >15 years old—a two-spot AFB microscopy should first be done. If one AFB is smear-positive, the patient should be treated with first-line TB drugs (FLD), whereas if the sputum is negative, antibiotic treatment should be followed by GenXpert mycobacterium tuberculosis or rifampicin (MTB/RIF) Assay. (2) For clients with a high risk to DR, HIV positives, children and presumptive TB involving the meninges, a GenXpert MTB/RIF Assay should be done. If MTB is detected but rifampicin resistance (RR) is not detected, the patient could be treated with FLD; if MTB and RR are detected, the patient should be treated as DR-TB and referred to a MDR-TB treatment center. But if MTB is not detected, a clinical re-evaluation such as a culture, drug susceptibility testing (DST) and chest X-ray should be done, and a GenXpert test should be repeated [7].

TB case finding can be passive or active. In Ethiopia, including the study area, TB case finding mostly rely on a passive case finding (PCF), which means the identification of persons who may have active TB at the health facility level and enhanced TB case finding at the community level. At the health facility level, health-care providers assess clients for signs and symptoms of TB, and start appropriate medical evaluations and diagnostic tests based on standard algorithms for TB. In addition, intensified case finding is used for high-risk patients such as HIV-infected individuals. In relation to enhanced TB case findings at the community level, health extension workers screen clients who come to the health posts with symptoms and conduct regular home visits for people with symptoms of TB. They also detect close a (household) contact of infectious TB patients, subsequently referring all TB people who screen positive for TB to nearby health centers for medical evaluation and further investigation [7]. By only relying on passive case finding, it is not possible to identify more than two-thirds of the estimated TB cases annually [7]. An active case finding (ACF) is a systematic and intensified screening of a latent TB infection in the community that attempts to identify individuals with TB earlier than they have previously been identified using a passive case finding [23]. An active case finding is more effective than a passive case finding, particularly in risk groups such as TB patients' household contacts, the homeless, prisoners and HIV-infected individuals [24]. PCF systems should be complemented with ACF strategies, especially for these just mentioned risk groups [24].

The TB prevalence identified with a combined active and passive case finding is much higher than with only a passive case finding [25]. Evidence shows that patients who came by

an ACF had less of a patient delay (the time delay from the onset of TB symptoms until the first visit to a health-care facility) and less total delay (the time delay from the appearance of the symptoms until the first diagnosis for TB) than those who had come by PCF. The total delay is the sum of the patient delay and health system delay [24].

The Jimma Zone (the study area) had a TB case notification rate of 47.1% in 2016 [26], which is much lower than the national level (67.3%), as well as the Oromia Region, where Jimma is located (65.4%) [4]. To the best of our knowledge, the causes for this low case notification rate are not yet explored, and there is no published study about barriers for a TB case finding in the study area. Moreover, community-based TB care in the study area is not yet scaled up to full coverage. To identify barriers is not adequate for solving the problem, while exploring the root causes seems crucial to tackling it [27,28]. The present study aimed to identify barriers and explore the origin of these barriers in relation to TB case finding in Southwest Ethiopia. This study is part of a larger project with the general aim of assessing the performance and quality of the current TB case finding and treatment services.

## 2. Materials and methods

### 2.1. Study setting and period

The study was conducted from August 2016 to January 2017 in the Jimma Zone including Jimma town, Southwest Ethiopia. Jimma is one of the zones in the Oromia Region in Ethiopia, and is located about 354 kilometers from Addis Ababa, the capital city of Ethiopia. In 2016, the Jimma Zone had a total of 17 woredas (districts) and two town administrations and comprises a total area of 18,412.54 square kilometers [29]. According to a 2007 census, it had a total population of 2,607,115, of which 90% were rural residents [30], with the projected population for 2016 being 3,174,418. In 2016, it had seven hospitals (five primary, one general and one specialized teaching and referral), 120 health centers and 494 health posts. The hospitals and health centers (health facilities) provide both diagnostic services and directly observed treatment for TB patients (DOT), whereas the health posts provide DOT for TB patients, in addition to TB screening and referrals to the health facilities. In the study area, TB diagnostic services coverage was 80%, and health posts' DOT coverage 23% [26]. Furthermore, non-governmental health institutions such as the Catholic mission and private clinics in Jimma Town also provide TB care. The Jimma Zone had approximately a 52% health service coverage (measured by the ratio of health facilities to the population) [29]. Communicable diseases are one of the major health problems in the study area [29]. The sources of funds and other resources for the implementation of the TB control program are the Ethiopian government, and global health agencies such as The Global Fund and The Center for Disease Control (CDC). Community-based TB care has started in many health posts, and there is a plan to scale up this type of decentralized TB care to all health posts or kebeles.

### 2.2. Sample size and sampling techniques

**2.2.1. In-depth interview.** Fifty DOT providers and program managers with experience of the diagnosis and treatment of TB, or the coordination of a TB control program for more than six months, were included with a purposive selection by using an intensity sampling technique [31]. We preferred this sampling technique to access information-rich individuals, and through that explore the entire picture of barriers to TB case finding. The included participants were 25 DOT providers, 23 local and two regional program managers. They were chosen because they represented TB control program implementers, and held relevant information about the TB control program in relation to their duties and responsibilities. In cases where two or more DOT providers were at one site, the more experienced provider was selected. In

addition, 10 TB patients under treatment were selected from five treatment sites for in-depth interview by using a "maximum variation" sampling strategy, which helped to obtain access to patients with different background characteristics, such as sex, age, level of education and phase of treatment [31]. The total sample size for the in-depth interview was 60 participants.

**2.2.2. Facility observation.** Eight woredas (districts) and one town administration were selected by a simple random sampling using the lottery method from 17 districts and two town administrations [32]. Subsequently, a total of 42 governmental health facilities, which comprised three hospitals, 11 urban health centers (located in districts or zonal towns) and 28 rural health centers, were selected from the sampled districts. Thus, different types of health facilities were included from a variety of districts to help obtain a comprehensive picture about the presence or absence of necessary infrastructure and other resources for TB case finding.

## 2.3. Study participants and data collection

We used different sources and methods for the data collection. Firstly, in-depth interviews were conducted with health professionals responsible for managing or coordinating a TB control program at different levels and providing DOT at the facility level, health extension workers who were providing DOT at the health post level and TB patients who were receiving their treatment (in the intensive and the continuation phase). The intensive phase treatment covers the first two months for new cases and the first three months for retreatment cases, whereas the continuation phase covers the last four months for new cases and the last six months for retreatment cases. For the interviews, a semi-structured interview guide was used (S1 Text). The interview guide was translated to the local language (Afaan Oromo) by an English teacher whose mother tongue is Afaan Oromo and re-translated to English by another English teacher whose mother tongue is Afaan Oromo.

The first author (BME) conducted the in-depth interviews using the local language to explore health workers' experiences and views about barriers for TB case finding in the study area. Different people were interviewed to access as many perspectives as possible in regard to the overall aim of the study. The patients were interviewed about their views and experience before and during the diagnosis for TB. The majority (55) of interviews were conducted at the health facilities. Based on their preferences, five interviews were conducted at patients' and DOT providers' homes. The study participants were assured of confidentiality and the interviews were conducted in a private setting. We tried our best to make the study participants became relaxed through small talk for the first few minutes, and to make sure that they were comfortable before the interview. The interviews lasted 60 to 90 minutes, and audio-taped and transcribed verbatim, with descriptive field notes written after each of the interviews. The interviews for the health workers were conducted as a form of conversations than questions and answers.

Secondly, the governmental health facilities were observed based on a predefined checklist to acquire an overview of the existing situation in relation to the availability of necessary infrastructure and other resources, such as electricity, human resources and reagents for TB diagnosis (S2 Text). It was conducted through observing the facilities' (TB room, laboratory room, drug store, outpatient department (OPD), patient waiting area, the overall infrastructure and environment), combined with asking questions to the responsible persons in the respective units of the facilities. The observation lasted 50 to 70 minutes for each facility. The facility observation was carried out by BME and a research assistant (health service management specialist). The data collection tools were prepared based on national and WHO guidelines of TB control programs, as well as existing studies [7,19,33–35].

## 2.4. Data analysis, data quality control and trustworthiness

A data analysis of the in-depth interview started during data collection by transcribing on a daily basis and by performing a preliminary analysis. The first author (BME) transcribed all the audio-taped interviews and reviewed the transcript by listening to the recordings. Descriptive notes were made during the review of the transcripts. This process also included the inclusion of new probing questions and a merging of some of the other questions in the interview guide. De-identified audio-taped interviews (transcripts) were reviewed by a health monitoring and evaluation specialist who is a native speaker of the local languages (Afaan Oromo and Amharic) for peer debriefing.

BME conducted the data coding and categorizing using atlas.ti version 7.1 software. The last author (MS) reviewed a list of codes with associated quotations and categories; BME and MS then discussed and agreed upon the final list of codes and categories. A thematic analysis approach was used for identifying, analyzing and describing patterns and deviant cases within the data. The data were summarized in major themes, such as inadequate resources for TB case finding, and subthemes like a shortage of human power, a limited basic infrastructure and a shortage of diagnostic equipment and supplies [36]. Lastly, we attempted to paint a broad picture of what all the study participants described as their views and experiences in regard to TB case finding. The data were presented through descriptions of events and experiences relevant to the aim of the study [37].

The facility observation data were checked and computed for frequency of presence and an absence of selected resources and stock-out of acid fast bacilli (AFB) reagents. Narrative data based on the observation of each health facility, and its environment was also written up. Consequently, the observations from the facilities were used to validate the findings from the in-depth interviews, in addition to enriching and deepening the understanding of the data.

We used a translated and pre-tested interview guide. Triangulating with both data sources and data collection methods helped to challenge the data from various sources and findings already found by other methods. The sample size was relatively large, with an extensive amount of fieldwork over five months. The in-depth interviews were conducted by putting an emphasis on collecting data sufficient to provide a description of the existing barriers. We stopped the recruitment of the study participants when we reached saturation. The data from different people at different levels helped to ensure a "thick description" of the different functions and experiences of people working within the TB control program, as well as the patients receiving the services [31,38]. The interviewer (BME) also tried her best to ensure that the relationship between her and the study participants was built on trust so that the participants felt safe.

The interviewer knew the local languages and culture, and she believed that she managed to establish an "optimal" distance (insider-outsider) from the study [31,38]. She considered herself as an insider due to her being a fluent speaker of the local languages, thereby helped her to understand the context. The outsider role was partly preserved by not working with the DOT providers and program managers. The research assistant had experience in doing qualitative interviews, and is a fluent speaker of the local languages. Finally, there were frequent discussions and consultations on the design, data analysis and result interpretations among the authors up to the manuscript approval. The purpose of the discussions was for experience sharing and to have a common understanding.

## 2.5. Ethical considerations

Ethical approval was obtained from The Regional Committee for Medical Research Ethics (REK), Norway with reference number of 2015/2124 REK sør-øst B and the Jimma University

Institutional Review Board, Ethiopia with reference number of RPGC/389/2016. Subsequently, permission was obtained from both regional and local health departments in Ethiopia. Additionally, the study participants were informed about the purpose of the study and written informed consents (oral for patients who could not read and write) were obtained. Furthermore, privacy and confidentiality were ensured by using numbers to identify and describe the study participants. The collected data were stored in a locked file cabinet (for a hard copy) and on a computer with a password (an electronic copy).

## 3. Results

### 3.1. Characteristics of the study participants

Of the 60 individuals participating in the in-depth interview, 50 (83%) of them were health workers, 42 (70%) were male and 44 (73%) were in the age range of 25–34 years. From 50 health workers, 23 (46%) of them had a first degree, and 25 (50%) of them had 2 to 5 years of work experience for their current position or responsibility (Table 1).

From the analysis of the in-depth interviews, two major themes and five subthemes were identified from the data. The major themes were: (1) Inadequate resources for TB case finding, and (2) Limited access to TB diagnostic services. These are described below in detail, with

**Table 1. Characteristics of the study participants for in-depth interviews in the Jimma Zone, 2016/7.**

| Variables | | Number |
|---|---|---|
| **Health workers (n = 50)** | | |
| Sex | Male | 38 |
| | Female | 12 |
| Age in years | $\leq 24$ | 9 |
| | 25–34 | 40 |
| | $\geq 35$ | 1 |
| Position or responsibility | DOT provider | 25 |
| | District level manager | 20 |
| | Zonal or regional level manager | 5 |
| Professional education | Certificate | 6 |
| | Diploma | 18 |
| | First degree | 23 |
| | Master (2nd degree) | 3 |
| Duration of experience for current responsibility | < 2 years | 16 |
| | 2 to 5 years | 25 |
| | > = 6 years | 9 |
| **Patients (n = 10)** | | |
| Sex | Male | 4 |
| | Female | 6 |
| Age in years | $\leq 24$ | 2 |
| | 25–34 | 4 |
| | $\geq 35$ | 4 |
| Duration of treatment | < 2 months (intensive phase) | 7 |
| | $\geq 2$ months (continuation phase) | 3 |
| Residence | Rural | 6 |
| | Urban | 4 |
| Type of TB | Pulmonary TB | 7 |
| | Extra pulmonary TB | 3 |

respective subthemes and illustrative quotes. It was supplemented by findings from the observations at the health facilities.

## 3.2. Inadequate resources for TB case finding

This theme has three subthemes: (1) shortage of health-care providers, (2) inadequate basic infrastructure, and (3) inadequate diagnostic equipment and supplies.

**3.2.1. Shortage of health-care providers.** A majority of the study participants among the DOT providers and program managers at various levels reported that there was a shortage of health-care providers, especially laboratory professionals for the provision of TB diagnostic services. They explained that sometimes they did not have trained laboratory professionals, committed DOT providers and health extension workers to provide quality TB services for their community. One participant explained:

*. [. . .] We do not have an adequate number of health extension workers and laboratory professionals [. . .].* (A district program manager)

In addition, our findings from observations at 42 health facilities show that there was an absence of trained laboratory professionals in 11(26%) of the facilities. These 11 health facilities were rural health centers which means there was a mal-distribution of the available professionals in the urban health facilities with rural health facilities.

The participants (program managers and DOT providers) described possible causes for the shortage of health-care providers. The shortage of laboratory professionals was claimed to be due to a closed laboratory program in some training institutions (Ethiopian politicians considering the saturation of laboratory professionals in the country). Other reasons related to a high turnover of laboratory professionals as many laboratory professionals leaving government institutions and joining private institutions. According to the participants, the shortage of health extension workers was due to health extension workers joining long-term training and not being substituted for. A lack of committed DOT providers was suggested to be due to an absence of risk allowance (incentive) and personal protective materials, such as masks for health personnel assigned to the TB clinic. Many underscored that working at a TB clinic has its own risk, as one participant expressed:

*DOT is very good if implemented strictly. Currently, health professionals do not want to be assigned at TB clinics. They are at risk: there are no personal protective materials, no risk allowance and necessary resources for TB.* (A DOT provider)

**3.2.2. Inadequate basic infrastructure.** A majority of the study participants of DOT providers and program managers stated that there was a limited basic infrastructure, such as water supply, electricity, proper roads and rooms. They described that they often did not have the required resources, such as separate and equipped TB rooms to provide quality TB services for their community. The majority of health posts (at community level) and several health centers did not have a clean water supply and electricity; most health centers did not have patient waiting area and sputum collection area. They often lacked proper roads and affordable transportation services in their catchment. One participant clarified:

*. [. . .]. Only five out of the nine health centers found in our district have electricity; four out of the nine health centers do not have electricity. Only four out of nine health centers have a clean water supply. In addition, our health posts do not kept minimum standards. [. . .].*

*Inconvenient roads also make it difficult to travel with a car during the rainy season.* (A district program manager)

We found an absence of a clean (protected) water supply in 13(31%) health facilities, an absence of electricity in seven (16.7%), an absence of room for TB treatment in five (11.9%), and an absence of a waiting area for patients in nine (21.4%) of the health facilities during the observation of the health facilities. From the 33 health facilities which had patient waiting area, only four (12.1%) of these had a separate waiting area for TB patients. Moreover, only one health center had a sputum collection area and only one hospital had a separate laboratory room for sputum examination.

Based on the interviews with the DOT providers, the consequences of an absence of the different resources were many. Patients expectorated just outside the laboratory rooms because of the absence of a sputum collection area, which could increase the transmission of TB to healthy individuals. Moreover, patients were travelling long distances on foot to access TB diagnostic services because of the absence of proper roads or affordable transportation services.

During our facility observation, we detected that many 10 (23.8%) of health centers had only two blocks and a shortage of rooms: one room divided into two rooms (for example, a TB room in front and a trachoma room in the back). Seven of the health centers' buildings were old and required maintenance, such as painting and reconstructing. In four health centers, staff were collecting rainwater during the rainy season because of a lack of pipe water. Additionally, nine of the patient waiting areas did not have adequate chairs or benches, while the waiting areas of three health centers were occupied with patient cards and non-functional tables.

The participants of program managers and DOT providers mentioned different possible causes for the shortage or absence of basic infrastructure. They explained that a lack or shortage of equipped and separated TB rooms was due to an inappropriate designing of the health facilities, as most newly constructed health centers had only two blocks. As a result, health personnel did not access separate TB rooms for TB-related services. A shortage or absence of a clean water supply, electricity and proper roads were explained as being related to budget constraint, a problem of the sustainability of projects (for water, electricity), as well as a low mobilization of the community for fund raising.

According to the participants, the budget constraints could be a result of inadequate budget allocation, inappropriate budget planning and a lack of accountants to use health-care financing system from internal revenue. Other causes, like an inappropriate utilization of resources such as money and an inadequate collaboration with other sectors, such as the education sector, were also mentioned. A participant elaborates on this issue:

*There is a low involvement of the community in contributing money for infrastructure. Even what was contributed in our kebele (village) was not used for our kebele [. . .]. Also there is a low engagement of the community to avail or contribute locally available resources.* (A DOT provider)

According to a majority of the study participants, an increased participation of the community in contributing money and availing locally available resources such as wood, stone and labor might help in the construction of protected spring water and for electricity.

**3.2.3. Inadequate diagnostic equipment and supplies.** The absence of functional microscopes and shortage of AFB reagents were major challenge for an early diagnosis of TB and follow-up, which was reported by many of the DOT providers and program managers. The

majority 39 (92.8%) of the health facilities had reagents for AFB (methylene blue, acid alcohol and carbol-fuchsin) at the laboratory room on the day of observation. However, the AFB reagents were not available in the drugstores of most health facilities: acid alcohol in 27 (64.3%), carbol-fuchsin in 26(61.9%) and methylene blue in 25(59.5%) health facilities on the day of observation. Moreover, the national TB control program guideline was not present in 28(66.7%) outpatient departments, 27(64.3%) laboratory rooms and 19(45.2%) TB rooms of the facilities. Functional weighing scales and masks were not present in 19(45.2%) TB rooms of the facilities. Nevertheless, these resources are supposed to be available in all 42 health facilities.

According to DOT providers and program managers, causes for the shortage of reagents and supplies were a result of a failure to timely distribute reagents and supplies for the health facilities. This was explained as being related to an inadequate number or total absence of vehicles (cars, motorcycles) to distribute the reagents and other supplies for the facilities. Almost all program managers mentioned that the absence of vehicles for TB control programs hindered them from conducting various activities. For example, they often could not deliver drugs and reagents in time or conduct a supportive supervision of the health facilities and health posts involved in providing TB care. They also mentioned other causes such as a delay of service providers in requesting the necessary resources, including reagents from district or zonal health offices, whereas many DOT providers reported that the reagents for AFB often reached the health facilities close to their expiration date (have short shelf life), which then expired within a short time. During the observations, this was confirmed as we found reagents for AFB with near their expiration date in the drugstores of some health facilities, and expired reagents in the drugstores of many health facilities.

## 3.3. Limited access to TB diagnostic services

This theme has two subthemes: These are the absence of nearby health facilities providing TB diagnostic services and health system delays in the diagnostic process.

**3.3.1. Absence of nearby health facilities providing TB diagnostic services.** Most participants, both patients and health personnel, spoke about how health facilities (hospitals and health centers) providing TB diagnostic services were far away from many of the patients. This was related to the use of time, as well as the indirect costs involved for transportation and examinations, such as x-rays and biopsy. In general, a long travel distance for patients due to the absence of nearby health facilities was reported as the main problem in relation to TB care. One patient, being in the intensive phase of treatment, talks about her struggle:

*. [. . .]. My residence is far away from this health center (a two-hour walk) and there is no nearby health post, the health post is also far away from my residence. [. . .].* (A TB patient)

The DOT providers and district level program managers also confirmed the problem with travel distance through several examples. A program manager spoke about one such example:

*.[. . .]. I have experienced a 50-year-old mother coming to a health center after walking for eight hours, I was shocked when I saw her. [. . .].* (A district level program manager)

One participant stated that his district had only seven diagnostic centers for TB out of 11 health centers found in the district, meaning that four health centers only provided TB treatment. Another participant described some of the problems related to this:

*. [. . .]. Our major problem is the absence of laboratory service in our health center, we are collecting sputum and sending it to other health centers for acid fast bacilli (AFB), and even we do not have a budget to do this.* (A DOT provider)

Most participants among the DOT providers and program managers expressed that it is important to provide TB services at health posts, including diagnostic services, so that everybody can access all the elements TB care. Some said that there might be patients in "hard to reach areas" dying of TB due to the topography of their districts, thus making TB diagnosis and care geographically inaccessible. For example, Sekachokerssa, Omonada and Dedo represent districts which are more difficult for a majority of people to seek healthcare due to their physical inaccessibility. However, a few district level program managers believed that all of their health centers and health posts were geographically accessible for their community members.

In the interviews, we explored what the program managers and DOT providers said to explain the possible causes for the paucity of decentralized sites providing diagnostic services in the study area. The majority of program managers and DOT providers mentioned the following causes: (1) an inadequate expansion of TB diagnostic sites to the community–most DOT providers and district level program managers mentioned that some of their health centers do not providing TB diagnostic services except the treatment; (2) the absence of proper road and public transportation services–in most rural areas the centers are only accessible by motorcycles. Even so, using motorcycles might be difficult during the rainy season; (3) the presence of hard to reach areas related to an inconvenient topography, which leads to physical inaccessibility for some patients; and (4) budget constraints.

**3.3.2. Health system delays in the diagnostic process.** We probed selected TB patients in the study area about their experiences in the process of the diagnosis and treatment of TB. A majority of the patients reported that they had been suffering from symptoms of TB, such as cough and chest pain for several months, before they were diagnosed as TB patients. Based on patients' descriptions, there seemed to be an unnecessary period of health system delay in obtaining the proper diagnostic service after the patients had sought such services at a health facility. The patients reported that it took between one to five months to recognize their illness as TB and start their treatment after they initiated contact with the health care providers. The delay was mainly related to a referral from health posts to health centers, as well as health centers to governmental hospitals for further investigation. One of the patients reported that he did not know what illness he suffered from before two months after his first health center visit; the health professionals simply gave him medicine (antibiotics) for seven days after a negative result of sputum examination. After two months without being diagnosed, he went to a private clinic which had an x-ray machine. Lastly, the patient was diagnosed after two weeks additional time as being pulmonary TB-negative with further investigation (with an x-ray).

Another patient reported that her illness was diagnosed as TB after five months of repeated visits to a health facility. Her diagnostic process was delayed due to a delayed sputum examination and chest x-ray. Yet, another patient reported that it took four months for his illness to be diagnosed, starting from the first visit in a health center. The patient was continuing to a private health facility where he was diagnosed for TB after different investigations like a sputum examination, blood tests and an x-ray, which he was not offered at the health center he visited. Subsequently, he was transferred from a private health facility to the health center he visited so that he could start the treatment, as only some private health facilities have permission to treat TB patients. However, he was not able to start the treatment immediately because the DOT provider was not there (at the health center). He was therefore again delayed, and started his treatment at the health center five days after receiving the diagnosis.

Some DOT providers and program managers spoke about laboratory service interruption, implying that no tests were performed for seven to 14 days in some health facilities. The interruption of laboratory tests could be one cause for the health system delay for diagnosis. This was solved by sending patients to nearby health centers for a sputum exam, and by sending laboratory technicians from one health center to another health center. The delay was described as having negative effects on the daily life of the patients, both in terms of suffering from symptoms of the illness such as cough and pain, as well as expenses related to TB investigations. Furthermore, the expenses might increase as a result of costs related to transportation and investigations done at private clinics. A patient expressed his feelings about how the delay had affected his daily life:

> *I think my daily life has been affected much during examination (before illness was known). I was examined at a private clinic with high expenses, although my illness was not detected at that time. I went to Jimma hospital (a referral hospital in Jimma) for further examination with an x-ray. During all this time, I was suffering from chest pain and sleeplessness; I couldn't sleep for several days. Moreover, I had many expenses and my job hours were affected. My workplace is out of this town and I have to travel about one hour in walking distance (to reach the clinic).* (A TB patient)

According to most patients, their daily life was highly affected due to the delay in the diagnostic process.

The origin for barriers of TB case finding and interrelated factors were summarized and presented by diagram (Fig 1).

## 4. Discussion

The present study explores patients', DOT providers', and program managers' views and experiences related to barriers for TB case finding in Southwest Ethiopia. These views and experiences are supplemented with findings based on observation of a variety of health facilities. Our findings suggest that there are different and interrelated barriers for TB case finding in the study area. Major barriers include inadequate resources, and limited access to TB diagnostic services. Furthermore, the origin and interrelation of the barriers were described [27,28].

Our findings suggest that there were inadequate resources which were necessary for TB case finding. The findings are consistent with previous studies in Ethiopia which reported that there were inadequate resources, such as trained health-care providers and laboratory reagents for the TB control programs [12,39,40]. Inadequate or the absence of necessary resources might significantly hinder the provision of TB diagnostic services as expected, and cause TB case notification to become low in the study area. Moreover, the shortage of these resources could influence the performance and quality of the TB control program.

According to the Stop TB partnership action framework for TB case detection, a limited basic infrastructure like clean water, a limited number of health workers and a lack of incentive systems for providers are all possible causes for a low case detection rate and a treatment delay [41]. A good and effective service provision requires a trained and competent staff serving with the correct medicines and medical equipment, which required adequate financing. An organizational system that provides proper incentives, such as an allowance to service providers and users, is necessary to obtain a desirable health outcome or for successful treatment. There is also a significant association between health workers' density and health service coverage and outcomes [42].

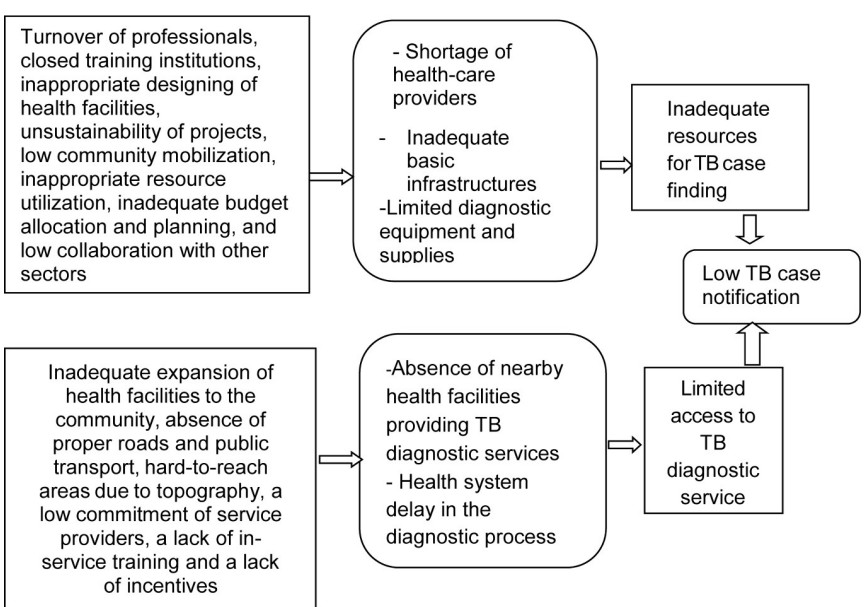

**Fig 1. Origin for barriers of TB case finding and interrelated factors, Jimma Zone, Southwest Ethiopia, 2016/17.**

Limited access to TB diagnostic services, being a result of an absence of nearby health facilities providing TB diagnostic services, was another barrier for TB case finding. Most of the rural residents of the study area do not easily access health centers and hospitals. As a result, individuals who have symptoms of TB are expected to seek TB diagnostic services by traveling long distance. Hence, the individuals might be delayed or not seek the services at all [12,16]. This could be related to the country's economy in availing TB diagnostic centers at the community level, and poor rural residents not able to afford costs related to transportation and investigations.

We found a long health system delay, ranging from one to five months. A much longer health system delay was observed in our study than in that of a study conducted in Gonder Town, Ethiopia, in which the median health system's delay was five days [5]. This difference might be due to the differences in study settings and methods. Our study included both rural and urban settings using a qualitative method, whereas the study in Gonder Town included only an urban setting and they used a quantitative method. As a result, rural residents might have less access to TB diagnostic centers and health information than urban residents. Accessibility to health services and awareness about TB might influence their early health-seeking behavior and follow-up after the first visit to the health facility. The health system delay could also amplify the total delay and TB transmission rate [43]. The health system's delay in the present study seems to primarily be related to a referral from a lower level to higher level health system for further investigations. This finding is similar to a previous study result in Ethiopia, in which the diagnostic delay was associated with care-seeking from multiple health care providers and visiting primary level health-care facilities for initial care [5].

Long distance traveling to obtain TB diagnostic services was another challenge for most patients from rural areas. Kleinman describe clients as a part of a community in a particular socio-cultural context, and health, illness and health-care related issues of communities as cultural systems. He divides the health-care system into popular, folk (traditional) and professional sectors [44]. The popular sector includes the family context of illness and care and social network, as well as community services related to the individual; the folk sector comprises

non-professional healing consultants and the professional sector contains modern medicine. A majority of the illnesses and healthcare, as well as most decisions on the time to seek help, where clients go and consult, and the way of compliance with treatments, are often dependent on the popular sector [44,45]. Consequently, limited access to health services due to distance could significantly affect the health-care seeking behavior of most of the rural population, which could lead to diagnostic delays and a low TB case notification. A long distance from the clinic was also reported in previous studies from many other countries and Ethiopia as a barrier for TB case finding by impacting their timely health-seeking behavior and lengthening a patient's delay [46–51].

TB case identification may be improved if the community could easily access health facilities with TB diagnostic services. Studies show that case notification and treatment outcomes increase in areas where community members have better access to facilities with TB diagnostic and treatment services [12,52,53]. Studies in Southern Ethiopia show that an involvement of health extension workers in sputum collection for microscopy at the community level significantly increased TB case finding and improved treatment outcomes [21,52]. Furthermore, TB case notification was higher in areas with better access to TB diagnostic and treatment facilities. Moreover, community-based interventions, such as active case finding played an important role for increased case notification rate [12]. A community-based active case finding using door-to-door household symptom screening, supplemented with laboratory tests, also increased TB case finding in rural China [54]. Moreover, active case finding through community outreach improved the speed of TB case finding, which indicated a possibility to reduce delays in TB diagnosis by at least half in Southern Ethiopia [20].

In Ethiopia, health extension workers (HEWs) are intentionally placed at the community level to reinforce the accessibility of the rural population to different health services, including TB care. The government case finding strategy also includes the identification of most rural presumptive TB cases in the duties and responsibilities of the HEWs [51,55]. This strategy could help in solving problems related to unmet health needs in general, and undiagnosed TB cases in particular, for the rural population.

In the present study, we recognized that the patients had difficulty to obtain even the simplest type of TB diagnostic service (AFB) due to absence of nearby health facilities with TB diagnostic services. A prompt TB diagnosis is crucial for interrupting TB transmission in the community. It is well known that the major source of TB spreading to healthy individuals is untreated AFB smear-positive TB patients. However, studies show that TB could be transmitted by AFB smear-negative TB suspects, because AFB smear microscopy has a very poor sensitivity and specificity compared to GenXpert MTB/RIF assay and MTB culture to detect Mycobacterium tuberculosis (MBT) [56,57]. A study revealed that from 168 AFB smear-negative sputum specimens, 28.57% and 34.52% were identified as MTB-positive by GeneXpert MTB/RIF assay and MTB culture, respectively [57]. In other studies, GeneXpert MTB/RIF assay had a higher sensitivity (93.75%) than AFB smear microscopy (50.00%) [56].

Other qualitative studies also revealed similar barriers to TB control programs, such as a shortage of human resources, inadequate access to TB services, and transportation problems, were reported along the Thai-Myanmar border [58]; a shortage of resources, poor accessibility and a capacity for TB diagnosis, as well as a weak community involvement, were reported in Northern Malawi [59]. Moreover, a shortage of resources, and a lack of access to TB diagnostic services at peripheral health facilities, were reported in Northwest Ethiopia [40].

According to the WHO's health system framework, there are six building blocks for strengthening health systems in order to improve health outcome. These include good health service delivery, a health workforce that performs well, a functional health information system, a strong health financing system, leadership and governance, as well as necessary medical

products, vaccines and technologies. All health systems should accomplish some basic tasks: the provision of services, having health workers and other essential resources, mobilizing and allocating finances, and maintaining of health system leadership and governance [42]. Our findings are related to four of the building blocks: limited access to TB diagnostic service delivery, an insufficient number and type of health workers, an inadequate budget and challenge to use the health-care financing system and a shortage of diagnostic equipment and supplies, including reagents. Ideally, all (the six) building blocks should be in place to improve health outcomes [42].

The roots of poor health outcome mostly seem to be based on a system failure related to a lack of well-established polices, process and procedures [60]. Most of the origins for the barriers of TB case finding in the present study also seem to be related to policies and process. To improve a specific outcome of health care within a health system, all the causes contributing to health outcomes should be known. After the origins of a particular health outcome are recognized, improvements can be done to help address these causes and finally change the outcome [61]. Hence, an improvement in the origins of barriers to TB case finding could increase TB case identification and improve the TB control program. Although some of the barriers are related to the country's poor economic condition and being difficult to tackle, other barriers can be improved through different interventions.

As a limitation, the present study reflects the views of a limited number of patients, DOT providers and program managers at different levels who participated in the study. We interviewed patients who have been on treatment, but not those individuals who had symptoms of TB in the community and had not yet sought TB care. Thus, we might miss essential information regarding barriers to TB case finding from symptomatic individuals who did not yet seek biomedical care. Moreover, the responses from the study participants might tend to be positive, and may not address their concern fully (social desirability bias) since most of the interviews were conducted at health facilities and health posts. We attempted to reduce this bias by informing about the objective of the study, assuring their confidentiality and indirect questioning, which meant asking about what others thought and felts. As a strength of the study, we used a relatively large sample size of in-depth interviews and facility observations. This helped to provide a great deal of data sources and method triangulation.

## 5. Implications for practice

We believe that the TB control program should address further decentralization of diagnostic services to the community. At the very least, all health extension workers should be trained and practice collecting sputum samples, preparing smears at the community or health post level and sending samples to laboratories for testing. This could have a positive effect on case finding, and may increase the case notification rate [51,55].

To help increase TB case identification in the community, it is better to learn from the Ivory Coast's process of decentralization for TB control. The Ivory Coast has carried out various activities, including the establishment of a steering committee, health site rehabilitation with lower costs by using containers, the integration of TB services, conducting dissemination workshops and the adoption of decentralization by stakeholders; and conducting training at different levels of health care providers, including traditional practitioners. Simultaneously, an ENGAGE-TB approach (integrating community-based tuberculosis care into the activity of non-governmental and other civil society organizations) was taken at the community level. These include a situation analysis, operational guideline development for community activities, community health workers' training and a high involvement of NGOs [62].

Furthermore, building future health facilities with TB diagnostic services should be done with a mind to separate potentially active TB patients from other sick patients by considering the location and proper design. Referral systems from a lower to a higher level should be strengthened with feedback to ensure that all referred patients have visited the clinic to which they were referred. Moreover, proper and timely distribution of TB diagnostic equipment and supplies from Zonal health departments or District health offices to the health facilities should be strengthened. A strong collaboration with different sectors and non-governmental organizations may help for the development of the necessary infrastructure's projects (water, electricity, roads, etc.) [42].

Active case finding using an outreach strategy could be an important strategy to enhance TB case finding within the community. The enhanced type of TB case finding may lead to an earlier diagnosis, and result in a reduced transmission of TB within the community. Active case finding through community-based chronic cougher follow-up have been shown to improve TB case finding and be helpful for socioeconomically underprivileged individuals in Ethiopia [63]. Attention should be given to improve TB case finding, and to reduce the perceived barriers, by making TB diagnostic services more accessible, affordable and acceptable by the community [22]. Identifying persons with symptoms and signs of TB, and the start of a medical evaluation as early as possible, could avoid the lengthy process and unnecessary delays. In addition, an innovative TB screening approach, such as screening any patient with a cough of any duration with at least one of the suggestive TB symptoms present, should be considered to improve TB case finding [64]. An integrated intensive case finding of TB, not only for HIV-infected individuals but also for patients with diabetic and cardiac problems, as well as maternal and child health clinics, could improve TB case finding [65].

It is possible to construct a TB patient waiting area and sputum collection area by mobilizing the community to avail or contribute locally available resources such as wood and stones, as well as free labor. Moreover, basic masks should be availed and provided not only for health workers who are providing TB care, but also for any coughing patients, to reduce nosocomial transmission. The role of social support systems like "Edir and Ekub" (community-based social support systems in Ethiopia) might help in providing psychological, financial and service support for individuals who have symptoms of TB. In addition, providing feasible incentives not only money but also moral support like education opportunities, reviewing performance and salary, could be possible solutions for a shortage of health-care providers and a low commitment of service providers [42]. These solutions may help to retain and motivate health-care providers in the governmental health system.

Finally, a future study using a community-based mixed method is necessary to identify further barriers of TB case finding in the community, and to improve community TB care.

## 6. Conclusion

There were different challenges in relation to TB case finding in the study area. Inadequate resources, and a limited access to TB diagnostic services, were major barriers identified in the present study. There is an interrelation between these barriers, and each barrier has its own basic causes. Strategies aimed at addressing the identified barriers for TB case finding should be recognized to increase TB case identification. These strategies could help to improve TB case finding in particular, and TB control programs in general. The findings of this study may help decision makers to focus on the origin of these barriers to help tackle the problems.

## Supporting information

**S1 Text. Interview guide and information sheet with consent form English and local language versions.**
(PDF)

**S2 Text. Checklist for facility observation.**
(PDF)

**S3 Text. Consolidated criteria for reporting qualitative research (COREQ) checklist.**
(PDF)

**S1 Data. In-depth interview data.**
(ZIP)

**S2 Data. Facility observation data.**
(SAV)

## Acknowledgments

We would like to thank the University of Oslo, Jimma University, Oromia Health Bureau, the Jimma Zone and Jimma Town health departments as well as all the study participants for providing us with the necessary support and information.

## Author Contributions

**Conceptualization:** Berhane Megerssa Ereso, Solomon Abebe Yimer, Christoph Gradmann, Mette Sagbakken.

**Data curation:** Berhane Megerssa Ereso.

**Formal analysis:** Berhane Megerssa Ereso, Mette Sagbakken.

**Funding acquisition:** Berhane Megerssa Ereso.

**Investigation:** Berhane Megerssa Ereso.

**Methodology:** Berhane Megerssa Ereso, Solomon Abebe Yimer, Christoph Gradmann, Mette Sagbakken.

**Supervision:** Solomon Abebe Yimer, Christoph Gradmann, Mette Sagbakken.

**Visualization:** Solomon Abebe Yimer, Christoph Gradmann, Mette Sagbakken.

**Writing – original draft:** Berhane Megerssa Ereso.

**Writing – review & editing:** Berhane Megerssa Ereso, Solomon Abebe Yimer, Christoph Gradmann, Mette Sagbakken.

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
