## [Decision Letter · Decision Letter 0]

24 Sep 2019

PONE-D-19-23348

Barriers for tuberculosis case finding in Southwest Ethiopia: A qualitative study

PLOS ONE

Dear Mr Ereso,

Thank you for submitting your manuscript to PLOS ONE. After careful consideration, we feel that it has merit but does not fully meet PLOS ONE’s publication criteria as it currently stands. Therefore, we invite you to submit a revised version of the manuscript that addresses the points raised during the review process.

We would appreciate receiving your revised manuscript by 30th October 2019. To enhance the reproducibility of your results, we recommend that if applicable you deposit your laboratory protocols in protocols.io, where a protocol can be assigned its own identifier (DOI) such that it can be cited independently in the future. For instructions see: http://journals.plos.org/plosone/s/submission-guidelines#loc-laboratory-protocols

We look forward to receiving your revised manuscript.

Kind regards,

Kwasi Torpey, MD PhD MPH

Academic Editor

PLOS ONE

Journal Requirements:

1. Please include a copy of the interview guide used in the study, in both the original language and English, as Supporting Information, or include a citation if it has been published previously.

Reviewers' comments:

Reviewer's Responses to Questions

**Comments to the Author**

1. Is the manuscript technically sound, and do the data support the conclusions?

Reviewer #1: Partly

Reviewer #2: Yes

2. Has the statistical analysis been performed appropriately and rigorously? 

Reviewer #1: N/A

Reviewer #2: N/A

3. Have the authors made all data underlying the findings in their manuscript fully available?

Reviewer #1: Yes

Reviewer #2: Yes

4. Is the manuscript presented in an intelligible fashion and written in standard English?

Reviewer #1: Yes

Reviewer #2: Yes

5. Review Comments to the Author

Reviewer #1: Summary:

The manuscript looks at a great deal of data collected from 60 in-depth interviews with health workers, program managers, and patients as well as clinic observations from 42 facilities to determine reasons for a low TB notification rate in a rural area of Ethiopia. The researchers have done a great deal of work to analyse and triangulate interview and observational data, showing where clinic resources fall short. On the whole, this is a well-researched manuscript and will provide a good addition to the qualitative literature on TB in low-resource settings once revisions are made to reduce repetition throughout the manuscript and strengthen the interpretation of findings.

Suggestions for major revision:

The manuscript could use a critical trim, as each of the sections have a lot of repetition. For example, the description of facility observation is described in multiple paragraphs of methods. In several places throughout the manuscript, two sentences will be used to say the same thing.

A second issue relates to one of the themes identified in the findings, namely 3.3. I am unconvinced by the data that there is a lack of information on TB in the community. Health workers often blame health delays on patients, whereas patients often recount long journeys to diagnosis once they make contact with the system. Often there is delay at both patient and facility level; however, the data presented on patient delay in your study is conjecture. Only 10 patients were interviewed – an unlikely number to reach saturation – and those interviewed did not report long delays between symptom onset and visiting a health facility. Both health workers and patients alike thought that more could probably be done to improve knowledge in the community, but the data reported does not support this finding. As such, I would advise removing the discussion on patient education and removing comparisons to other studies in which patient delay was noted – this is unrelated to your study, and you acknowledge the lack of data from patients not reaching clinic as one of your study limitations.

Furthermore, I would like to push the authors a bit further in terms of discussion and recommendations. To add substantively to the qualitative literature, it is insufficient to recommend more resources and more education when resources are limited and there is a plethora of research to show that more education does not necessarily lead to changes in practice. Your interview findings on resource shortages are important and are well backed by the clinic observation. The next step is to place these findings in context for the reader, e.g. what are the particular circumstances and potential ways to improve the situation within the existing context and resources. Does the existing data include health worker perspectives on why there is such high staff turnover (not just conjecture from program managers)? Money and risk aversion are not the only motivators. It would be interesting to see some perspectives from the DOT workers on what motivates them to stay or prompts them to move on, if such data is available.

Furthermore, while the data is compelling to show that there is a lack of quality decentralised services in the region of study, it would be useful to discuss models of decentralisation in low-resource settings that could be implemented in the Ethiopian context. It also may not be reasonable to expect a lab technician to be available to conduct AFBs in all clinics, but building a better supply chain despite poor roads and long distances has been successful in other settings. Rebuilding existing centres may not be economically viable, but building future clinics with a mind to separating potentially active TB patients from other sick patients may be an important recommendation, as well as identifying other low-cost strategies that have been used elsewhere to ensure safety of other patients in waiting rooms and during sputum collection – for example having patients go outside to a private area to bring up sputum and having chairs available outside the clinic or providing basic masks for any coughing patients while in existing waiting areas to reduce nosocomial transmission (although this also requires a degree of sensitisation to prevent stigma). Try to find examples of these in the literature to provide practical recommendations, or tie your findings to existing theories in new ways that will help policymakers and providers think of new ways to approach the issues identified.

Minor suggestions and editorial corrections:

Introduction:

It would be helpful to have an overview of correct TB management in the Ethiopian context. In South Africa, for instance, sputum is first sent for GeneXpert when PTB is suspected, whereas reference is made to stock outs of AFB. It would help the reader to understand the basic TB testing algorithm in use during your study.

The use of DOT and DOTS are at times confused throughout the manuscript. DOT refers to the act of directly observing TB treatment, whereas DOTS refers to the programme, which involves not just the observation of treatment, but supply chain management, testing facilities, and political will, etc. Be careful not to mix these up when referring to frontline DOT workers versus those who make up other aspects of the DOTS strategy.

Be cautious of the language used to describe patients. The use of terms like “TB suspects” and “defaulting” imply wrong-doing or blame, when TB transmission and loss to follow-up are more systemic issues. For example Line 74: “TB suspect cases” should be changed to “People who screen positive for TB” and in Line 80: “Treatment default rate” is more appropriately referred to as loss to follow-up of those on treatment.

Line 62-63: quality of TB care at the primary care level is another (needs a period)

Line 93: they also detecting close contact of (incorrect grammar) / please define close contacts

Line 95: remove the ‘s’ from passive case findings

Line 95-98: This sentence is long and confusing. Consider revising

Line 106-107 repeat definition, plus the bracketed information is incorrect - it is time from first symptoms to diagnosis, as stated previously. Please delete

How were notification and case rates determined?

Line 109: unpublished annual report – annual reports should be referenced appropriately

Methods:

Line 121-22, the number of districts and administrative towns should not be referenced as an observation.

Line 124: “In 2016 it has” change ‘has’ to ‘had.’

Line 127 The phrase “TB suspect identification” should read “TB screening”

Line 129, what is the definition of coverage being used?

Line 133: Since the CDC is a government organisation, I would suggest replacing NGO with global health agencies.

Line 137: The correct terminology is purposive selection (not purposely selected).

Line 164 “Secondly, a facility observations” should read “Secondly, facility observations” (remove ‘a’)

As Table 1 describes demographics, it would normally be placed under the results section

Section 2.4 uses quotations around qualitative research concepts such as “thick description” and “optimal” distance as insider-outsider to the data. These concepts should be referenced from the literature.

Results:

Line 248-250, it would read more clearly if the authors stated that “three main themes were identified from the data: 1, 2, 3,” rather than listing the themes in brackets. Although a personal preference, I also try to avoid the use of phrase “themes emerging from the data,” as this seemingly undermines the difficult task of the researchers whose work it is to identify the themes after a lengthy analysis process.

Line 340: add an ‘s’ to causes: “They also mentioned other cause”

Line 350: replace “told” with “spoke about”

Line 351: replace “was” with “were”

Lines 375-76 This sentence is confusing in the use of “few”. Perhaps it would be clearer to say “to explain the paucity of decentralised sites providing diagnostic services”

Section 3.2.1: There is a difference between having access, i.e. existence of a nearby health facility and gaining access, i.e. accessibility as in feeling safe and supported at health facilities. You refer to this in the discussion. Did this difference come up in any of the patient interviews?

Discussion:

In response to 3.2.1: Recently there has been a larger focus placed on the quality of TB care once reaching the facility (versus access), i.e. health workers thinking to test for TB as well as tests being affordable and available. The continued use of AFB, which has very poor sensitivity compared to GeneXpert is also of concern. It is important to touch on these aspects in the discussion.

Lines 463-465: Referring to what works in Sweden, a high-income, low-burden country is not necessarily transferable to the Ethiopian context. Are there other more relevant references in this regard?

Line 529: First use of the acronym HEWs. Use consistently throughout or not at all.

Reviewer #2: Dear Academic editor,

Thank you for inviting me to review the manuscript, “Barriers for tuberculosis case finding in Southwest Ethiopia: A qualitative study”

I reviewed the manuscript and have the following comments for the authors to improve their manuscript.

General comments

In the abstract section

1. Results: “Inadequate resources for TB case finding”; this is a general term and the authors should describe the resources that are inadequate for TB case finding briefly.

2. ......inaccessibility to TB diagnostic facilities...... this should also be clearly described because accessibility/access has various dimensions. Is this physical inaccessibility, affordability, acceptability or availability?

3. Conclusion:..... “Limited resources for TB case finding”..... ‘Inadequate resources for TB case finding’ in the results section

Introduction

4. The authors should check the use of full stop they used it before the references. It was not consistent. For instance, on line number 44 and on others, the full stop was used before references.

5. The authors used 2015’s data or report; it would have been better to use the recent reports of TB case notification rates of Ethiopia for global comparisons.

6. Line number 123, why the authors could not put the number of health posts? They reported approximately 500 despite knowing the number of all kebels.

7. The health service coverage of the Zone reported to be approximately 52%. Is this based on physical coverage or health facility density to population ratio? The total population of the Zone was 2,607,115, the number of health facilities was also known; the authors should check the population to health centre density or other facilities as well; 2, 607115/123. One health centre is expected to serve 25000 people. Moreover, DOTs and TB diagnostic services coverage should also be described.

Methods

8. Table 1 under “Duration of experience for current

Responsibility” the authors mentioned only <2years, 2-5 years and 9-10 years; why the authors didn’t mention those who had 6-8 years and above ten years of experience?

9. Line 189, “ A relaxed atmosphere”.... this is not clear for a new reader and I would suggest using another term.

10. Line number 189 “frequent discussions and consultations among the authors”..........did not specify why they did the discussion and consultations; the discussions were about what....?

11. Line numbers 231 and 234-235; are the research assistant and interviewer different or are the same people? This is not clear

Results

12. In line numbers 265-269 the authors mentioned different reasons for shortage of laboratory technicians. One of this was the “due to a closed laboratory program in some training institutions (Ethiopian politicians considering the saturation of laboratory professionals in the country)” .... what about the problems in maldistribution of the available human health workforce. Did the authors see for evidence of urban concentration and rural scarcity? This could also affect the availability and contributes to inequities which in turn affect TB control performance. In some rural settings even if the professionals are available in the job market budget constraints to hire the professionals could be one of the challenges.

13. Under “inadequate basic infrastructure”..... it would be better if the authors mention the numbers or proportions instead of ...... “ The majority of health posts”.......... “several health centers”.... “most health centers”........

14. Discussion: the strengths and limitation of the study design as well as possible bias and measures taken to reduce the bias were not discussed. The authors should consider this in their discussion

6. PLOS authors have the option to publish the peer review history of their article (what does this mean?). If published, this will include your full peer review and any attached files.

Reviewer #1: Yes: Jody Boffa

Reviewer #2: Yes: Mesay Hailu Dangisso,MPH, PhD

---

## [Author Response · Author response to Decision Letter 0]

31 Oct 2019

First of all, we would like to thank the Academic Editor for his careful reading, and the Reviewers for their high quality and constructive reviews of our manuscript. Sincerest thank you so much for your prompt response on our manuscript. 

We have modified the manuscript in response to the extensive and insightful reviewers’ comments. 

Reviewer 1 – We have carefully considered and addressed all of your comments and suggestions in the revised manuscript. These were very helpful. Thank you so much. 

Reviewer 2- We have carefully considered and addressed all of your comments and suggestions in the revised manuscript. These were very helpful. Thank you so much. 

Academic Editor - We have carefully considered and included the requirements commented by you in the revised manuscript and submission system. Thank you so much for your valuable comments.

---

## [Editor Report · Decision Letter 1]

4 Nov 2019

PONE-D-19-23348R1

Barriers for tuberculosis case finding in Southwest Ethiopia: A qualitative study

PLOS ONE

Dear Mr Ereso,

Thank you for submitting your manuscript to PLOS ONE. After careful consideration, we feel that it has merit but does not fully meet PLOS ONE’s publication criteria as it currently stands. Therefore, we invite you to submit a revised version of the manuscript that addresses the points raised during the review process.

We would appreciate receiving your revised manuscript by 30th November 2019. To enhance the reproducibility of your results, we recommend that if applicable you deposit your laboratory protocols in protocols.io, where a protocol can be assigned its own identifier (DOI) such that it can be cited independently in the future. For instructions see: http://journals.plos.org/plosone/s/submission-guidelines#loc-laboratory-protocols

We look forward to receiving your revised manuscript.

Kind regards,

Kwasi Torpey, MD PhD MPH

Academic Editor

PLOS ONE

Additional Editor Comments (if provided):

Thanks for the revised manuscript. There are a couple of issues that requires attention

1. Rebuttal to reviewers' comment: Your point by point rebuttal fails to show exactly what was done and where in the manuscript (Section/Page/Line or paragraph) that the change was effected. Please ensure the revised rebuttal letter captures this level of detail

2. Please thoroughly copyedit the manuscript possibly using a native speaker. There are several language errors observed

Thank you

---

## [Author Response · Author response to Decision Letter 1]

20 Nov 2019

Reviewer 1. We have carefully considered and addressed all of your comments and suggestions. 

 Thank you so much for your valuable comments and suggestions.

Reviewer 2. We have carefully considered and addressed all of your comments and suggestions. 

 Thank you so much for your valuable comments and suggestions

---

## [Editor Report · Decision Letter 2]

25 Nov 2019

Barriers for tuberculosis case finding in Southwest Ethiopia: A qualitative study

PONE-D-19-23348R2

Dear Mr Ereso,

We are pleased to inform you that your manuscript has been judged scientifically suitable for publication and will be formally accepted for publication once it complies with all outstanding technical requirements.

With kind regards,

Kwasi Torpey, MD PhD MPH

Academic Editor

PLOS ONE
---

## [Editor Report · Acceptance letter]

18 Dec 2019

PONE-D-19-23348R2 

Barriers for tuberculosis case finding in Southwest Ethiopia: A qualitative study 

Dear Dr. Ereso:

I am pleased to inform you that your manuscript has been deemed suitable for publication in PLOS ONE. Congratulations! Your manuscript is now with our production department. 

With kind regards,

on behalf of

Professor Kwasi Torpey 

Academic Editor

PLOS ONE